# Improving the Ability of Deep Neural Networks to Use Information from Multiple Views in Breast Cancer Screening

**Nan Wu**[1]                             nan.wu@nyu.edu
**Stanisław Jastrzębski**[2,1]               staszek.jastrzebski@gmail.com
**Jungkyu Park**[2]                           jp.park@nyu.edu
**Linda Moy**[2,3,4]                     Linda.Moy@nyulangone.org
**Kyunghyun Cho**[1,5,6]                     kyunghyun.cho@nyu.edu
**Krzysztof J. Geras**[2,3,1]                       k.j.geras@nyu.edu

[1] *Center for Data Science, New York University*

[2] *Department of Radiology, NYU Grossman School of Medicine*

[3] *Center for Advanced Imaging Innovation and Research, NYU Langone Health*

[4] *Perlmutter Cancer Center, NYU Langone Health*

[5] *Department of Computer Science, Courant Institute, New York University*

[6] *CIFAR Associate Fellow*

## Abstract

In breast cancer screening, radiologists make the diagnosis based on images that are taken from two angles. Inspired by this, we seek to improve the performance of deep neural networks applied to this task by encouraging the model to use information from both views of the breast. First, we took a closer look at the training process and observed an imbalance between learning from the two views. In particular, we observed that layers processing one of the views have parameters with larger gradients in magnitude, and contribute more to the overall loss reduction. Next, we tested several methods targeted at utilizing both views more equally in training. We found that using the same weights to process both views, or using modality dropout, leads to a boost in performance. Looking forward, our results indicate improving learning dynamics as a promising avenue for improving utilization of multiple views in deep neural networks for medical diagnosis.

**Keywords:** Breast cancer screening, deep neural networks, multimodal learning, multiview learning.

## 1. Introduction

Breast cancer screening decreases mortality by enabling early detection of cancer (Autier et al., 2012). In the screening process using a mammogram, two views of the breast are taken: bilateral craniocaudal (CC) and mediolateral oblique (MLO) (Figure 1). These two views capture the breasts from above and from the side, respectively. Using both views in breast cancer screening has demonstrated to be essential to make an accurate diagnosis (Gur et al., 2009). In practice, radiologists usually consider a finding more plausible if it is visible in both views.

Deep neural networks (DNNs) have shown promise in aiding interpretation of breast cancer screening exams (Kyono et al., 2019; Shen et al., 2020; Schaffter et al., 2020; Geras

et al., 2019; Wu et al., 2018). However, despite the critical importance of utilizing information contained in both views, relatively little attention has been paid to this aspect in deep learning-based approaches to breast cancer screening. Importantly, it is not self-evident that DNNs utilize information contained in both views, even when they are designed to process both CC and MLO views simultaneously, such as the late-fused multiview networks trained end-to-end and adopted in literature (Wu et al., 2019a; McKinney et al., 2020; Geras et al., 2017). In a related problem of learning using multimodal data, e.g. speech and sound, it is a common phenomenon that DNNs fail to utilize information in all modalities (Wang et al., 2019).

Inspired by how radiologists read mammogram images, we seek to improve DNNs for this task by encouraging them to utilize information in both views. Our first contribution is a study on the training dynamics of a multiview network. In particular, we show that in the multiview network, the parts of the model operating on the MLO view contribute more to the overall loss reduction than the other parts operating on the CC view. Meanwhile, significantly larger gradient norms are observed for the parameters of the layers only associated with the MLO view throughout training. We hypothesize that this causes a form of overfitting, where the model learns to rely too strongly on the MLO view.

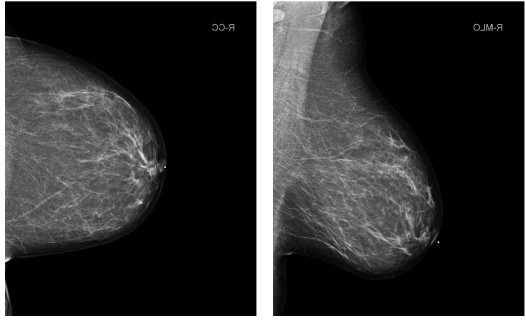

Figure 1: The standard screening mammography views. Left: bilateral craniocaudal (CC), right: mediolateral oblique (MLO).

Our second contribution consists of comparing different methods for utilizing both views, inspired by the literature on multimodal learning. We observe that two different methods, including modality dropout (Neverova et al., 2015), improve performance of the above studied multiview network.

## 2. Related work

Multimodal learning aims to build models that can process, and relate information from multiple input modalities (Baltrušaitis et al., 2018). Typical applications include combining visual and audio signals for content understanding (Neverova et al., 2015), and object recognition from visual observations of multiple views, sometimes also called multiview learning (Jia et al., 2019; Su et al., 2015; Wang et al., 2015).

Attempts to utilize multiple views in breast cancer screening with DNNs can be traced back to Carneiro et al. (2015), who trained models on MLO and CC views separately, then used the features from the last fully connected layer to train a multinomial logistic regression model. Motivated by Su et al. (2015), end-to-end trained multiview DNNs were proposed by Geras et al. (2017). Recently, late-fused multiview DNNs are commonly adopted in research on breast cancer screening, including experiments with other techniques from multimodal learning literature, such as pretraining (Carneiro et al., 2017; Kyono et al.,

2019), weight sharing (McKinney et al., 2020; Wu et al., 2019a) and attention (Shachor et al., 2019).

Despite the trend of building networks that learn from multiple views jointly end-to-end, it is not self-evident that multiview DNNs utilize information in both views optimally for the breast cancer screening task. A recent investigation shows that cancers visible only in one view are more often seen in the CC view than the MLO view (Korhonen et al., 2019). It is still not clear how this difference between the two views influences the performance of the multiview DNNs. As pointed out by Wang et al. (2019), training with multiple modalities (views) jointly under a single optimization strategy can be sub-optimal when models overfit and generalize at different rates in learning from different modalities. Wu et al. (2019a) mentions that a DNN that processes both views of the breast separately outperforms a DNN that processes them simultaneously. However, in breast cancer screening, radiologists make decisions by fusing information from both the MLO and the CC view. Motivated by the importance of using both views in the clinical practice, we delve deeper into understanding and improving using both views in deep learning for breast cancer screening.

## 3. Data and task

We conducted experiments with a dataset of 229,426 breast cancer exams (1,001,093 images) from 141,472 patients (Wu et al., 2019b). In each exam, there are images for both left and right breasts. We treat each breast as an instance and do not differentiate between left and right breast in training and inference. Across the entire data set (458,852 breasts), malignant findings were present in 985 breasts (0.21%) and benign findings in 5,556 breasts (1.22%). All findings were confirmed by at least one biopsy performed within 120 days of the screening mammogram. Images for the two views (CC and MLO) of the same breast share the same label.

The original mammogram images are cropped into the shape of 2677×1942 pixels for the CC view and of 2974×1748 pixels for the MLO view, before being passed as inputs to the model. Besides the mammogram image, we provide two "heatmaps" as extra channels in the inputs to the model. These "heatmaps" are generated by a classifier trained with small mammogram patches, and are used in the "image-and-heatmaps" model in Wu et al. (2019a).

We split the data into training, validation, and test sets. We trained the model for two tasks: predicting the absence/presence of malignant and benign findings in the breast. Following Wu et al. (2019a), we use benign prediction task as an auxiliary task

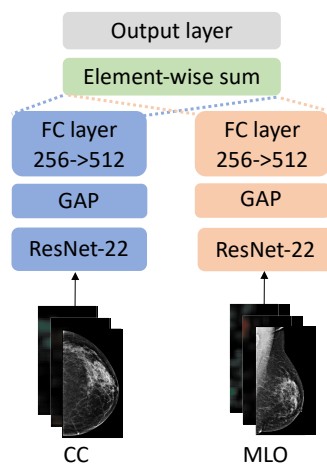

Figure 2: The network architecture used in the paper. Both views are processed by a separate ResNet-22 column, global average pooling layer (shorten as GAP), and finally a fully connected layer. The two resulting representations, $h_{cc}$ and $h_{mlo}$, are fused $(h_{cc} + h_{mlo})$ as input to the output layer, consisting of two binary classifiers predicting the presence or absence of malignant and benign findings.

to regularize the model and only consider the perfor-
mance on the malignancy detection task during model selection and evaluation.

## 4. Experimental setup

We use an architecture similar to what was in Wu et al. (2019a), shown in Figure 2. Each view is processed by a separate ResNet-22 column[1], followed by global average pooling, and a fully connected layer that maps the 256-dimension vector into a 512-dimension representation. The two resulting representations ($h_{cc}$ and $h_{mlo}$) are merged in the end by element-wise summation as $h_{cc} + h_{mlo}$. Predictions for benign and malignant tasks are made by the output layer consisting of two independent binary classifiers operating on the above merged representation. The model is trained end-to-end to minimize the sum of the two cross entropy losses, denoted as $\mathcal{L}$. This is a common practice in multimodal learning called late fusion (Ngiam et al., 2011; Baltrušaitis et al., 2018). We name this model as "Joint ResNet" and conduct analysis to understand possible challenges in utilizing multiple views with this architecture.

The training procedure largely follows Wu et al. (2019a). We downsample the negative class to balance the distribution in each training epoch. Specifically, samples used in each training epoch consist of: 1) all exams with followup biopsy records; 2) same number of exams without any biopsy record, randomly sampled from the training set. We use the Adam optimizer (Kingma and Ba, 2015) with a minibatch size of four. We use AUC to measure models' performance for malignancy prediction. We save the best checkpoint of the model according to the AUC it achieves on the validation set and report the AUC it reaches on the test set.

## 5. What makes using both views of the breast difficult?

Our goal is to improve the ability of our model to utilize information in both views of the breast. Unfortunately, naive approaches to using multiple views in breast or lung cancer detection networks tend to work poorly (Wu et al., 2019a; Bertrand et al., 2019). In Wu et al. (2019a) the best performing model simply averages predictions made based on separate views. For lung cancer detection, Bertrand et al. (2019) concludes that *"using the PA [frontal posteroanterior] and lateral views jointly doesn't trivially lead to an increase in performance but suggest further investigation"*.

What makes using both views of the breast difficult? As a first step we must try to diagnose the problem. We took inspiration from multimodal learning, a closely related setting where the input to the network consists of two or more modalities (e.g. sound and speech). Wang et al. (2019) have noticed that multimodal learning is challenging due to the fact that parts of the network associated with different modalities tend to train with different speeds. This might lead to a form of overfitting where the output of the network is determined mostly by one of the modalities.

---

1. ResNet-22 is a varaint of ResNet (He et al., 2016), proposed in Wu et al. (2019a) to handle high resolution images specifically.

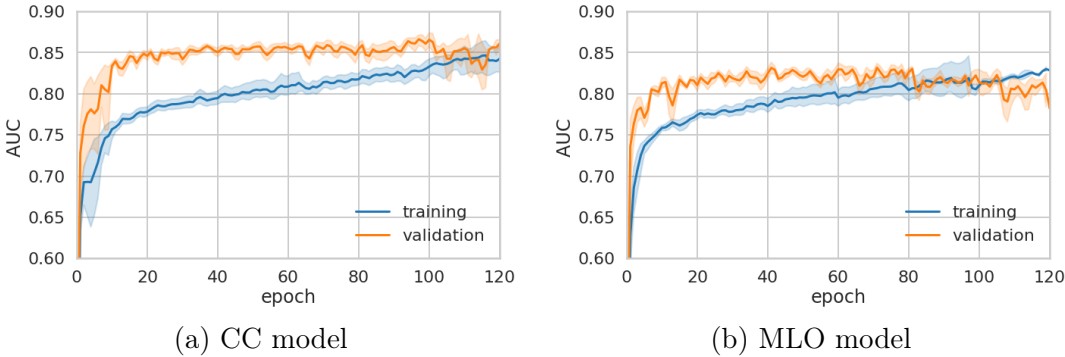

(a) CC model                              (b) MLO model

Figure 3: The training (blue) and the validation AUC (orange) of DNNs using only the CC view (a), and DNNs using only the MLO view (b). Models using only the MLO view overfit earlier and achieve worse generalization performance on the validation set. The shaded areas represent a standard deviation across ten models trained using different learning rates. The AUC on the training set is lower than the AUC on the validation set because we downsample negative examples in the training set Experimental setup.

To investigate if a similar phenomenon occurs in our setting, for each of the views (CC and MLO), we trained two uni-view counterparts of the above multiview model consisting of the ResNet-22 column, the fully connected layer, and the output layer.

We trained 10 models for each view with learning rates sampled from $[10^{-5}, 3 \times 10^{-3}]$ at logarithm scale. Figure 3 shows the learning curves for both groups of models. In general, the MLO models achieve their best validation AUC earlier than the CC models. Furthermore, the best performing CC model achieves an AUC of 0.864 on the test set while the AUC achieved by the MLO model is 0.789.

Related observations were made in Wang et al. (2019) in the context of multimodal learning. Similarly, our results show that the information contained by the two views is different in the sense that it leads to different learning speeds and different final performances of the models.

Ultimately, we are interested in understanding what makes training a model using both views difficult. To investigate the training of the multiview DNN (see Figure 2), we grouped parameters of the model into three parts, $\theta^{shared}$ from the output layer, $\theta^{CC}$ and $\theta^{MLO}$ from each ResNet-22, and the following fully connected layer. We investigated the importance of each group of weights for the overall training dynamics with two metrics.

First, we studied the Euclidean norms of mini-batch gradients (gradient norm) for $\theta^{CC}$ and $\theta^{MLO}$, i.e. $\|\frac{\partial \mathcal{L}}{\partial \theta^{CC}}\|$ and $\|\frac{\partial \mathcal{L}}{\partial \theta^{MLO}}\|$, respectively. Gradient norm has been related to the training speed in the literature. For example, the Adam optimizer (Kingma and Ba, 2015) utilizes gradient norm to regulate the magnitude of the updating steps. In multitask learning, Chen et al. (2017) use gradient norm to equalize training speed between different tasks. Figure 4(a) shows that $\|\frac{\partial \mathcal{L}}{\partial \theta^{MLO}}\|$ is larger than $\|\frac{\partial \mathcal{L}}{\partial \theta^{CC}}\|$ for most of training iterations. This suggests that the MLO column is more important for the overall loss reduction.

To further investigate this we use Loss Change Allocation (LCA) (Lan et al., 2019). LCA quantifies how much each parameter contributes to the overall loss reduction. Summing the scalars computed with LCA over all elements of $\theta^{MLO}$ and $\theta^{CC}$, we can compare their

relative importance for the overall loss reduction. We performed LCA for the first 5 epochs (we only looked at the early phase of training due to a large computational cost of LCA) and reported the cumulative loss changes contributed by $\theta^{CC}$ and $\theta^{MLO}$ in Figure 4(b). Analogously to the gradient norm, we observed that loss change contribution measured with LCA for $\theta^{MLO}$ is significantly higher than for $\theta^{CC}$.

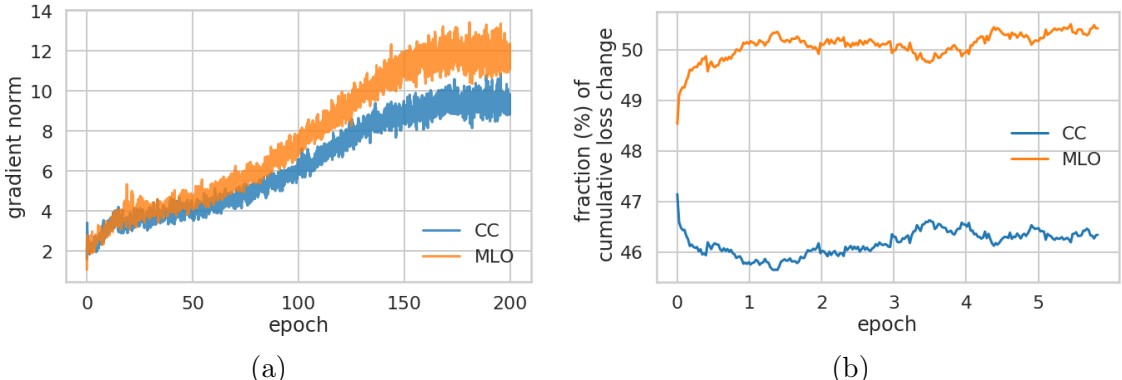

(a)                                    (b)

Figure 4: Figure (a) compares the gradient norms of parameters between the MLO (orange) and CC (blue) columns. Figure (b) shows contributions to loss change measured using Loss Change Allocation (LCA) for both sets of parameters. The figures show that the weights of the MLO column have a larger gradient norm for most of training, and contribute over 50% of the loss reduction in the first 5 epochs. These two observations suggest a form of overfitting: the output of the model depends too strongly on the MLO view.

To summarize, we observed that when trained separately, a model that uses only the MLO view trains faster at the beginning and overfits faster. Consistently, we observed a larger gradient norm and a larger contribution to loss change for the MLO column in the multiview model. Thus, this training procedure seems to be sub-optimal considering that when trained separately the CC model achieves better generalization. Assuming similar observations hold, generally, we hypothesise that *the difference in training speed encourages a form of overfitting where output of the model depends too strongly on only one of the views, and it is a key factor contributing to the difficulty of using multiple views in deep neural networks applied to medical diagnosis.*

## 6. Improving utilizing information in both views of the breast

Building on the intuition developed in the previous section, in this section we compare several methods for improving the ability of the model to utilize information in both views. We begin by describing the tested methods, which we organise into two groups: model variants, and regularization techniques. We treat Joint ResNet model shown in Figure 2 as the baseline.

### 6.1. Model variants

We tested the following two variants of Joint ResNet.

**Shared ResNet**   Perhaps the most natural way to encourage using information in both of the views is to share weights between the MLO and the CC columns. Specifically, we used a single ResNet operating on MLO and CC view to generate representations for the two fully connected layers. A related approach was used by McKinney et al. (2020). We will refer to this model as "Shared ResNet".

**Split ResNet**   We also tested a similar approach to one used in Wu et al. (2019a) to process information from the MLO and the CC view. We trained two uni-view models simultaneously, one for each of the views, with a loss calculated on the averaged output predictions of the two models. We will refer to this model as "Split ResNet".

### 6.2. Regularizers

We tested the following four regularization methods. While most of these methods can be applied in conjunction with any model, for simplicity we focus on studying the effect of applying them to the Joint ResNet.

**Pretraining**   A common approach in the multimodal learning literature is to apply the transfer learning technique (Neverova et al., 2015; Kyono et al., 2019). It consists of two stages: 1) pretraining, where we train separate models for each of the modalities; 2) fine-tuning, where we initialize each column in the multiview network with the weights from the corresponding uni-view model, and further train the multiview network with all modalities. Here we considered two variants: fine-tuning the entire model, and fine-tuning only the output layer. We find the latter to perform better and use only this variant in the rest of the paper. We will refer to this regularizer as "Pretraining".

**Modality Dropout**   Modality Dropout is a technique developed for multimodal learning (Neverova et al., 2015). To discourage the model from relying too strongly on one of the modalities, Neverova et al. (2015) proposes to use a dropout mask such that it completely masks input coming from one of the modalities. To adapt it to our model, we dropped the 512 dimensional output of each of the two FC layers randomly, before passing it to the fusion (summation) module. We considered the dropout rate of each view as a hyperparameter and sample it independently and uniformly from $[0, 1)$ for each view.

**Split Learning Rates**   Inspired by our analysis, we investigated another approach. We proposed to use separate learning rates for different components of the model to counteract the differences in training speed between the columns. We will refer to this modification as "Split Learning Rates".

   More precisely, we split the Joint ResNet in the same way as in the above section (What makes using both views of the breast difficult?) into three components: $\theta^{shared}$, $\theta^{CC}$ and $\theta^{MLO}$. We sampled numbers independently and set them as learning rates for the resulting components.

**Gradient Blending**   Finally, we investigated "Gradient blending", a technique proposed recently by Wang et al. (2019) for multimodal learning. Based on similar observations to the one we made in the previous section, they propose to calibrate training speed across

modalities through a weighted loss of the form

$$\mathcal{L} = \sum_i w_i \mathcal{L}_i + w_{joint} \mathcal{L}_{joint},$$

where $\mathcal{L}_i$ is the loss of a classifier that only uses input from one input modality, $\mathcal{L}_{\text{joint}}$ is the loss of a classifier that uses all modalities, and $w_i$ and $w_{\text{joint}}$ are scalar weights. Adapting their approach to our setting we arrive at

$$\mathcal{L} = w_{cc} \mathcal{L}_{cc} + w_{mlo} \mathcal{L}_{mlo} + w_{joint} \mathcal{L}_{joint},$$

where $\mathcal{L}_{joint}$ is the original loss used while the other two terms, $\mathcal{L}_{cc}, \mathcal{L}_{mlo}$, are two additional losses that we compute by adding two additional heads that read features outputted by the two ResNet-22 columns. Each head consists of a fully connected layer, and an output layer.

To tune $w_{mlo}$ and $w_{cc}$ we followed a similar procedure to Wang et al. (2019), which is to estimate the training speed of each model trained separately with each of the modalities as well as the joint model. Due to the class imbalance in our dataset, we measured the speed of overfitting and generalization based on cross-entropy loss and AUC, rather than accuracy. We have included more details in Appendix C.

**Other experimental details** We trained each model for 100 epochs and three repetitions to regulate the noise introduced by random initialization and randomization in data sampling. For models in the "Modality Dropout" group, we extended the training to 150 epochs given the slower convergence speeds. For "Pretraining" experiments, in which we freeze the two ResNets, we shortened the training to 50 epochs.

We conducted a moderate hyperparameter tuning for each group to ensure the fairness of the comparison. Details on the model selection are presented in the Appendix B.

We reported the test performance of the model which achieved the highest validation performance (averaged over the three repetitions) within each group. Following Wu et al. (2019a), during testing we augmented each example 10 times, by sampling the size and the location of the cropping window, and reported the AUC calculated with the model's average prediction of the 10 runs. The overall inference time for a model is about 8.6 hours on a single NVIDIA Tesla V100 GPU.

### 6.3. Results

**Comparing different architectural changes** Table 1 summarizes the results for the different architectural variations we considered as well as the uni-view models. For each group, we report the results from a single network (mean and standard deviation across three repetitions), and from an ensemble of the three repetitions, denoted as "3x ensemble". We can draw two main conclusions from these results. First, Split ResNet performs worse than uni-view model trained on CC views only. It further corroborates that using information in both views is challenging and not all joint training strategies will lead to a success.

Second, sharing weights between the two ResNet achieves the best AUC of 0.879 compared to the AUC of 0.872 achieved by Joint ResNet. This serves as an additional justification for sharing weights between ResNet reading different views (Wu et al., 2019a; McKinney et al., 2020).

Table 1: The test AUC of uni-view models and different model variants.

|  | AUC | 3x ensemble |
| --- | --- | --- |
| uni-view MLO | $0.789 \pm 0.010$ | 0.802 |
| uni-view CC | $0.864 \pm 0.010$ | 0.874 |
| Split ResNet | $0.854 \pm 0.016$ | 0.866 |
| Joint ResNet | $0.872 \pm 0.005$ | 0.887 |
| **Shared ResNet** | **$0.879 \pm 0.003$** | **0.890** |

Table 2: The test AUC of Joint ResNet models with different regularization techniques.

|  | AUC | 3x ensemble |
| --- | --- | --- |
| Joint ResNet | $0.872 \pm 0.005$ | **0.887** |
| + Pretraining | $0.830 \pm 0.012$ | 0.836 |
| + Split Learning Rates | $0.864 \pm 0.008$ | 0.878 |
| + Gradient Blending | $0.870 \pm 0.014$ | 0.880 |
| + **Modality Dropout** | **$0.876 \pm 0.009$** | 0.886 |

**Comparing different regularizers**   Next, we compared the effect of applying different regularizers to Joint ResNet. Table 2 summarizes the results. The main observation is that using Modality Dropout improves the AUC over the baseline. Joint ResNet with Pretraining generalizes poorly on the test set, showing this is not a helpful technique for at least this specific dataset.

In summary, among the methods we compared, Shared ResNet and Modality Dropout improve the performance of the baseline model. To examine the stability of this observation, we ran experiments on networks that do not use heatmaps as input (see Section Data and task). We compared on this setting Joint ResNet, Shared ResNet, and Joint ResNet with Modality Dropout.

Table 3: The test AUC of models trained without heatmaps.

|  | AUC | 3x ensemble |
| --- | --- | --- |
| Joint ResNet | $0.694 \pm 0.046$ | 0.692 |
| + Modality Dropout | $0.702 \pm 0.065$ | **0.779** |
| Shared ResNet | **$0.713 \pm 0.032$** | 0.728 |

Results for models using only the images, without the additional heatmaps as inputs, are presented in Table 3. We observe that the above conclusions transfer to this setting in the sense that both identified methods which improve performance over the baseline.

## 7. Conclusions

In this paper, we first analysed what makes using multiple views difficult and made a connection with the training dynamics. According to gradient norm and LCA, we observed that training was largely dominated by weights specific to the one of the views (MLO). We hypothesized that this caused the model to rely too strongly on the MLO view.

Using these insights, we investigated how to better utilize information from both views of the breast within the training of a multiview network. We ran our experiments on a model that achieves a performance close to radiologist-level (Wu et al., 2019a). We identified two methods that boost performance: (1) sharing the weights between the subnetworks applied to different views, (2) using modality dropout that masks one of the views out with a certain probability during each training step.

We examined a wide range of techniques adopted in multimodal learning literature, and perhaps surprisingly many methods ("Pretraining", "Split learning rates", and "Gradient blending") did not improve the performance. The methods that did boost performance (sharing weights between the ResNet-22 columns or using modality dropout), led to relatively modest improvements.

In summary, while we identified techniques that improved performance of the model, we conclude that improving the ability of deep neural networks to use information from multiple views it still a largely open research question. We propose that improving the training dynamics is a particularly promising direction for the future.

## Acknowledgments

The authors would like to thank Catriona C. Geras for correcting earlier versions of this manuscript, and Mario Videna, and Abdul Khaja for supporting our computing environment. We also gratefully acknowledge the support of Nvidia Corporation with the donation of some of the GPUs used in this research. This work was supported in part by grants from the National Institutes of Health (R21CA225175 and P41EB017183).

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

## Appendix A. Detail of experiments for Loss Change Allocation

The LCA framework is computationally expensive to use. We changed the number of samples in each epoch and estimated the loss change on samples used in the current epoch, rather than computing for the entire training set. In addition, instead of applying the algorithm per minibatch, we added a step size of $k$ and estimated the loss change: $\mathcal{L}_{(}t) - \mathcal{L}(\theta_{t-k})$ as a replacement for $\mathcal{L}_{(}t) - \mathcal{L}(\theta_{t-1})$. In each epoch, we sampled 970 exams from the entire training set, which is 10% of both positive and negative cases compared with the original sample size used to form each training epoch. We perform LCA at each 60 iterations of minibatches. We trained the model with a learning rate of $10^{-5}$. Figure 5 shows the overall loss change and the error stands for the difference between the actual loss change and the estimated loss change after allocated on each parameter with the algorithm. We recorded the LCA for approximately six regular training epochs. As we observed, the major loss change happens at the first few epochs and the contribution of each component, especially their relative relationship will not change significantly in the following training period.

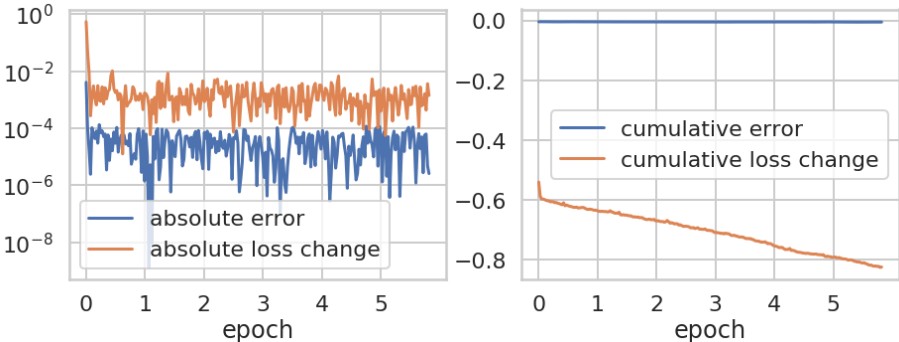

Figure 5: Error between the actual decrease in loss and the LCA metric for the baseline model with a learning rate of $10^{-5}$.

In Figure 6 and Figure 7 we present gradient norms and LCA recorded when training the models with different learning rates. When we change learning rates, the trend of gradient norm changes accordingly, which is consistent with previous observations in the literature (Jastrzębski et al., 2019). There is always a difference between the gradient norm of columns on CC and MLO view. MLO column tends to have higher gradient norm at the early phase. For LCA, column on MLO dominates the contribution to loss change across the training for all three models with different learning rates.

## Appendix B. Experimental details for model selection.

For "Joint ResNet", "Shared ResNet", and "Split ResNet", we sample ten numbers from $[10^{-5}, 10^{-3}]$ at logarithm scale as learning rates and train ten sets of models, each of three repetitions varying in random seeds. The learning rates for the best performing set among the ten are $6.9 \times 10^{-5}$, $4.62 \times 10^{-4}$, and $4.4 \times 10^{-5}$ for "Joint ResNet", "Shared ResNet", and "Split ResNet', respectively.

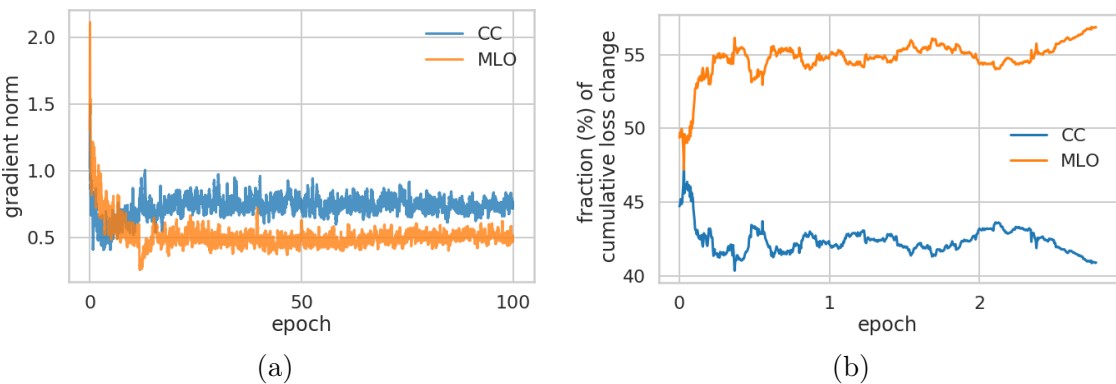

(a)            (b)

Figure 6: Gradient norm and LCA when training Joint ResNet with a learning rate of $10^4$. Figure (a) compares the gradient norm of parameters between the MLO (orange) and CC (blue) columns. Figure (b) shows Loss Change Allocation (LCA) for parameters in CC and MLO columns.

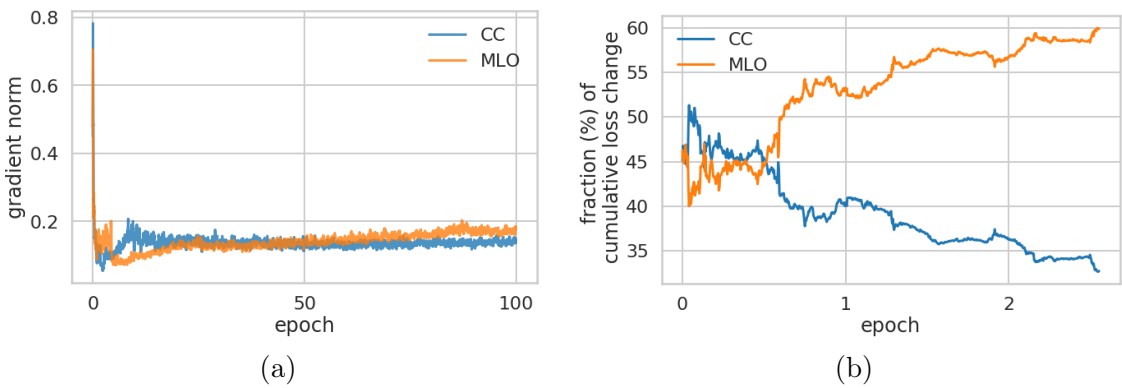

(a)            (b)

Figure 7: Gradient norm and LCA when training Joint ResNet with a learning rate of $1 \times 10^3$. Figure (a) compares the gradient norm of parameters between the MLO (orange) and CC (blue) columns. Figure (b) shows Loss Change Allocation (LCA) for parameters in CC and MLO column.

For "Pretraining", we initialize the model columns with parameters from the corresponding best performing uni-view models. In the fine-tuning stage, we follow the schema for the above variants and train ten model sets varying in learning rate.

For "Modality Dropout", we train ten model sets with a learning rate of $1 \times 10^{-5}$ and another ten with a learning rate of $1 \times 10^{-4}$. For each model set (consisting of three repetitions), we sample numbers uniformly and independently from $[0, 1]$ as the dropout rates for CC and MLO views. We spend more resources on the search of dropout rates rather than learning rates to enable us to focus on the unique effect of "Modality Dropout" as a regularizer on the training dynamic between CC and MLO view.

Under similar intuition, for "Split Learning Rates", we adopt the following schema. First we set $1 \times 10^{-5}$ and $1 \times 10^{-4}$ as base learning rates, denoted as $\eta$, each for ten model sets. Then we sample, $\alpha^{CC}, \alpha^{MLO}$, and $\alpha^{shared}$, independently and uniformly from $[-1, 1]$

and obtain learning rates for each model components: $\eta^{CC}, \eta^{MLO}, \eta^{shared} = \eta \times 10^{\alpha^{CC}}, \eta \times 10^{\alpha^{MLO}}, \eta \times 10^{\alpha^{shared}}$. The resulting learning rates lie in the range from $1 \times 10^{-6}$ to $1 \times 10^{-3}$. Distributions of the sampled learning rates for each components are presented in Figure 8

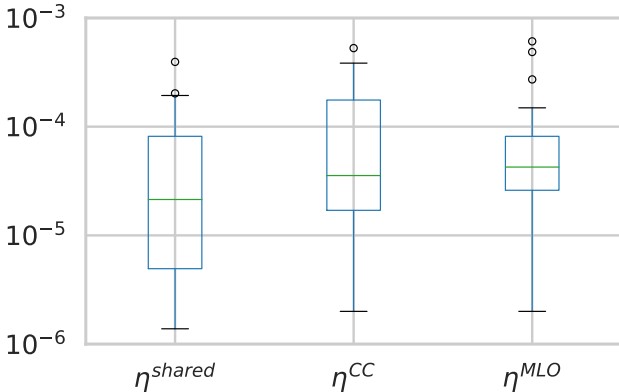

Figure 8: Distribution of learning rate for each model components, $\eta^{CC}, \eta^{MLO}, \eta^{shared}$.

## Appendix C. Experiments with Gradient blending

We train two uni-view models and the multiview model following the offline algorithm from (Wang et al., 2019) with global learning rates of $10^{-4}$. We calculate $w_{cc}$, $w_{mlo}$ and $w_{joint}$ based on AUC and loss on the validation set and we average weights for each model over specific window of epochs (from 0 to 100 epochs or from 90 to 100 epochs) as estimated weights used in the loss to train the multi-heads model:

$$\mathcal{L} = \hat{w}_{cc}\mathcal{L}_{cc} + \hat{w}_{mlo}\mathcal{L}_{mlo} + \hat{w}_{joint}\mathcal{L}_{joint},$$

We list the four sets of weights collected in different manner in Table 4. All models are trained with a fixed learning rate of $10^{-4}$ and with three repetitions. Among the 4 models trained with different weights combinations, we report the test performance of the one achieves the best AUC on validation set, averaged over the three repetitions.

Table 4: Estimated weights for gradient blending.

| metric | window of epochs | $\hat{w}_{joint}, \hat{w}_{cc}, \hat{w}_{mlo}$ |
|---|---|---|
| **loss** | $[0, 100]$ | 0.367, 0.300, 0.333 |
| | $[90, 100]$ | 0.373, 0.286, 0.341 |
| **AUC** | $[0, 100]$ | 0.319, 0.398, 0.283 |
| | $[90, 100]$ | 0.446, 0.498, 0.056 |

In addition, we conduct an ablation study on the gradient blending algorithm. Except for estimating the weights on the full validation set, we use 10% of the validation set instead

and use average weights from 0 to 100 epochs in the gradient blending loss. Besides, we train one more model with swapped weights on CC and MLO. We also train a model with equal weights on the three components of the loss. Results are listed in Table 5. We do not observe significant differences on their performance with the original weights. Surprisingly, the model trained with equal weights achieves the best AUC.

In addition, we observe high volatility in the weights estimated with this algorithm. Given the relatively higher fluctuation of the AUC on validation set, it can be hard to use this method to find representative weights to calibrate the training for our task.

Table 5: Performance of gradient blending models with different weights.

|  | $\hat{w}_{joint}, \hat{w}_{cc}, \hat{w}_{mlo}$ | **AUC** |
|---|---|---|
| **equal weights** | 0.333, 0.333, 0.333 | 0.870 |
| **loss** | 0.340, 0.325, 0.335 | 0.853 |
| swapped* | 0.340, 0.335, 0.325 | 0.858 |
| **AUC** | 0.246, 0.122, 0.632 | 0.868 |
| swapped* | 0.246, 0.632, 0.122 | 0.868 |

