# OpenReview forum: "Improving the Ability of Deep Neural Networks to Use Information From Multiple Views in Breast Cancer Screening"
_MIDL.io/2020/Conference — MIDL 2020_

### Official Review · AnonReviewer3 · 2020-03-13
**Improving the Ability of Deep Networks to Use Information From Multiple Views in Breast Cancer Screening**

**Rating:** 4
**Confidence:** 5
**Recommendation:** Oral

**Summary:**

This paper utilizes two images captured at two different angles for breast cancer diagnosis. In particular, images from both the views are fed to the Resnet-22 separately and later, fusing of FC layer is being done, followed by, two binary classification predicting presence or absence of malignant and benign findings.

**Strengths:**

Multi-view information fusing is state-of-art technique to improve the classification performance in medical imaging. Results are being presented using different variants of  multi-views information fusing.

**Weaknesses:**

I find the results to be compared and validated with state-of-art techniques for breast cancer diagnosis. This would be interesting to see where the paper is standing in terms of existing research. Discussion of results should be more elaborate.

**Justification Of Rating:**

Paper is well written and easy to follow. It is combining muliple views together to have better classification performance. Comparison of proposed method with some recent methods in the literature is needed before acceptance.

**Paper Type:**

both

**Questions To Address In The Rebuttal:**

1) What is a sense of checking gradient norms of views while picking one of them for further processing?
2) Why Resnet-22? Which layers are freezed of this network during training?
3) Comparison of results with the recent work in breast cancer diagnosis should be presented.
4) What is the overall inference timing during testing?
5) Results of only considering one view at a time should be compared.
6) Why original image is being resized to 2677x1942 bigger size. Wouldn't it increase overall computation time?
7) Are the heatmaps carry some information for cancer diagnosis? Please elaborate with appropriate citations.
8) What if one average or stack FC layers of both views? How it will impact the performance?
9) One discussion separate section is needed in the manuscript.

**Special Issue:**

no

---

> ### Author Response · Authors · 2020-03-28
> **Response to Reviewer #4**
>
> Thank you for your time and the comments. We followed your suggestion to revise the paper, especially a discussion section to further elaborate the results and the limitations. We address specific questions below.
>
> ''1) What is a sense of checking gradient norms of views while picking one of them for further processing?''
>
> Could we ask for clarification on what do you mean by “picking one of them for further processing”? Assuming that you refer to models regularized with multimodal dropout, we fully agree that in this case the significance of gradient norms might not be clear. However, we do not perform such a comparison in the paper.
>
> ''2) Why Resnet-22? Which layers are freezed of this network during training?''
>
> In the paper, we largely follow experimental choices made in Wu et al. (2019) In particular, ResNet-22 is a backbone architecture used in Wu et al 2019 for high resolution images. None of the layers are frozen during training, except for models in the “Pretraining” group, where we freeze the two ResNet-22 columns. We hope this answers the question. We will also add a clarification to the paper.
>
> ''3) Comparison of results with the recent work in breast cancer diagnosis should be presented.''
>
> Comparison with prior work is challenging due to the lack of standardized benchmarks in the field. We compared the AUC achieved in our paper with comparable models that utilized the same dataset in the discussion section of the revised paper.
>
> ''4) What is the overall inference timing during testing?''
>
> The inference time the model needs to iterate over the test data set is 3100 seconds on average with a single NVIDIA Tesla V100 GPU. We do ten iterations with data augmentation to evaluate each model. The overall inference time is about 8.6 hours.
>
> ''5) Results of only considering one view at a time should be compared.''
>
> Thanks for the suggestion. We added results for the model trained with one view at a time in the revised version.
>
> ''6) Why original image is being resized to 2677x1942 bigger size. Wouldn't it increase overall computation time?''
>
> Following Wu et al, we used the original image sizes (which were cropped to a smaller size to cover the most of the breast), i.e. we did not upscale the image. The primary reason for using the images in the original resolution is that fine details are crucial for detecting malignant changes in the breast. Perhaps the best example are microcalcifications. These are small changes in the breast tissue that are visible on the mammogram as bright pixels, which would be much less visible if we did not use the original image size.
>
> ''7) Are the heatmaps carry some information for cancer diagnosis? Please elaborate with appropriate citations.''
>
> We follow Wu et al. here. Heatmaps are generated by the patch-level classifier as proposed in by Wu et al. They contain pixel-level probability predictions of malignant/benign findings. Results in Wu et al. suggest that the information in the heatmaps is highly predictive of cancer; models in Wu et al. that use heatmaps strongly outperform models that do not. We will clarify this in the revision.
>
> ''8) What if one average or stack FC layers of both views? How it will impact the performance?''
>
> These are interesting questions. We will add experiments exploring different operations for merging the two views in the camera-ready version.

---

### Official Review · AnonReviewer4 · 2020-03-13
**Interesting study on how to help a classifier use multiple views more efficiently**

**Rating:** 3
**Confidence:** 4
**Recommendation:** Poster

**Summary:**

In this paper, the authors show that classifiers trained using multiple views or modalities tend to rely too strongly on one or the other of the input branches.

The hypothesis is that this problem occurs because each branch of the model learns at a different speed and contributes to the training loss at a different scale.

To address this issue, the authors investigate different ways of training the model and different regularizers.
The experimental results show that weight sharing among the different branches and modality dropout are boosting the performance of the classifier.

**Strengths:**

I enjoyed reading the paper. It is very well written and structured.

The context and issue are clearly stated.

The hypothesis is well defined and backed by experiments.

The solutions investigated are interesting, particularly the choice of regularizers.


**Weaknesses:**

Heatmaps are also given as input to the model. How are they influencing the model performance? An experiment with only the images would be interesting to see.

Training the classifier to predict begnin findings is described as an auxiliary task for regularization. It would be good to have a baseline experiment where the classifier is trained only for the main task to see the influence of this regularization.

The model variants section could be extended. In Figure 2, the fusion of the 2 branches is done with an element-wise sum. As the problem addressed is the fusion of the information extracted in the different branches, it would be good to investigate other fusion alternatives (feature concatenation, product).


**Detailed Comments:**

Suggestions:

In Tables 1 and 2, it would be better to sort the methods by increasing AUC.

To compensate for the different learning speeds, instead of varying the learning rates, have you tried training the branches with different numbers of training iterations? For example, training the CC branch 5 iterations before training the MLO branch 1 iteration?


**Justification Of Rating:**

The paper is well structured and the pain point and hypothesis are well explained and illustrated by experiments. The proposed solutions are meaningful. The experimental part could be extended a bit but the results are already interesting.

**Paper Type:**

methodological development

**Questions To Address In The Rebuttal:**

Could you give some details on the training for the Shared ResNet variant? Are you updating the model in 2 steps, one for each branch?

You also mention test time data augmentation. Could you give some details about the data augmentation used?

Apart from that, any elements addressing the weaknesses listed above are welcome.

**Special Issue:**

no

---

> ### Author Response · Authors · 2020-03-28
> **Response to Reviewer #3**
>
> Thank you for your time and suggestions on the paper.
>
> We acknowledge the limitations in the experimental scope of our work. The main reason for not including more models or ablation studies in this version of the paper is the computational cost, due to the number of the images in the dataset (1,001,093) and their high resolution. We train each single model for at least 100 epochs to let it fully converge which takes more than a week using an Nvidia Tesla V100 GPU. We chose to include heatmaps to accelerate the training and to improve accuracy since models trained with the original images only do not perform well without pretraining on screening BIRADS classification (Wu et al., 2019).
>
> Having said that, we already started running the suggested experiments, including models trained without heatmaps, models trained with heatmaps but without the benign classification task as regularizer and models with different fusion alternatives. We will add them to the camera-ready version. In particular, we agree with the reviewer that the choice of the fusion operator might be important for how well the model utilizes information in both views.
>
> In what follows, we address the remaining questions and suggestions.
>
> ''Could you give some details on the training for the Shared ResNet variant? Are you updating the model in 2 steps, one for each branch?''
>
> For the Shared ResNet, we tie weights for the two ResNet-22 columns, which is equivalent to using a single ResNet-22  as a feature extractor and obtaining representations for CC and MLO views. We keep two separate fully connected layers to transform each representation into 512 dimensions before applying the summation. We will clarify this in the revision.
>
> ''You also mention test time data augmentation. Could you give some details about the data augmentation used?''
>
> We crop from the original image with a window with a size of 2677x1942 pixels for the CC view and 2974x1748 for the MLO view. For data augmentation, we slightly change the size of the cropping window and resize the cropped image using bicubic interpolation to fit the desired size for the model. We also add noise around the chosen location of the window. All types of noises are kept within 100 pixels in all directions. We will clarify this in the revision.
>
> ''In Tables 1 and 2, it would be better to sort the methods by increasing AUC.''
>
> Thank you for this suggestion. We will change this in the revision.
>
> ''To compensate for the different learning speeds, instead of varying the learning rates, have you tried training the branches with different numbers of training iterations? For example, training the CC branch 5 iterations before training the MLO branch 1 iteration?''
>
> This is an interesting idea. We will add this as a separate regularizer in the camera-ready version of the paper.

---

> ### Comment · AnonReviewer4 · 2020-03-31
> **Response to Rebuttal**
>
> Thanks to the authors for the clear answers in the rebuttal.
>
> Dear AC,
> I found the answers in the rebuttal convincing and would like to change my rating to strong accept.
> Thanks.

---

### Official Review · AnonReviewer2 · 2020-03-13
**Interesting topic, and good paper, not too many results**

**Rating:** 3
**Confidence:** 4
**Recommendation:** Oral, Poster

**Summary:**

This paper explores the idea of classifying breast X-ray images to find tumors by using two different image orientations at the same time. This is a very interesting topic in DL-based medical image processing, therefore perfectly fits in the scope of the conference, because human radiologists also consider images taken from multiple orientations. We are not sure how to integrate the knowledge from X-rays of multiple orientations, therefore the kind of research this paper is presenting is very important. Unfortunately, this paper is making only very limited steps towards achieving its ambitious goal.

**Strengths:**

The topic is very interesting. Integration of knowledge from multiple image orientations is definitely something that we need to use. Not just in this, but many other imaging applications.
The authors present a clear and comprehensive review of previously published methods, which is very valuable for readers.
The experiments are presented in details, mostly in the appendix.
The amount of training and testing data is very impressive.

**Weaknesses:**

The results and conclusions of the paper are very limited. The difference between experimental tests is small. The conclusions and lessons learned are limited.
However, the authors could make an effort to extend the paper with a Discussion section. As usual in scientific papers. Currently, some thoughts that would belong to the Discussion are mixed in the Results and other sections.
A link to your source code would be even more useful than the appendix.

**Justification Of Rating:**

The topic is very relevant and interesting. The authors are well prepared and present the literature accurately. The only reason I'm not recommending accept super strongly is that the results are not very interesting. Probably due to the difficult problem the authors trying to solve.

**Paper Type:**

methodological development

**Questions To Address In The Rebuttal:**

I think you should reorganize the text and make a Discussion section. An important part of Discussion would be the limitations of your research.

**Special Issue:**

yes

---

> ### Author Response · Authors · 2020-03-28
> **Response to Reviewer #2**
>
> Thanks for your time and the remarks about the paper.
>
> In this paper, we utilized a few different methods, including gradient norm and Loss Change Allocation, to investigate the challenges in using multiple mammogram views with DNNs. We evaluated a range of methods on a clinically realistic dataset.
>
> We acknowledge the methods improve not significantly from the baseline model, the Joint ResNet. As far as we know, such comprehensive comparison focusing on multimodal techniques, especially on a dataset with 1,001,093 images, is missing in the literature of deep learning with breast cancer screening. We believe our work can provide useful insights for related research topics and also prove inspirations for developing better deep learning techniques for breast cancer screening.
>
> Thank you for your suggestion for extending the Discussion section. We incorporated your remarks and provided an enriched discussion on the limitations of our experiments in the revised paper. We will also release the code with the final version of the paper.

---

### Official Review · AnonReviewer1 · 2020-03-14
**Incremental comparison study**

**Rating:** 2
**Confidence:** 4

**Summary:**

The manuscript evaluates ways for the training of networks integrating information from multiple-view mammography images for the classification of malignant and benign cases. Due to the discrepancies in the CC and MLO views, training of the sub-networks could suffer from uneven training gradient which causes suboptimal results. Different integration strategies and regularization strategies are compared.

**Strengths:**

The manuscript addresses a legitimate issue in training networks integrating multi-modality data and the experience learned from the study has value guiding similar practices in such training tasks.
Latest techniques in the filed are identified, cited, and compared.

**Weaknesses:**

Since the overall approach is largely inspired by previous publications (Wu et al., 2019; McKinney et al., 2020) the novelty of the posed network is limited.
Also the proposed model architecture which makes slight modification of existing models fails to outperform previously published results.
More details need to be provided, e.g. whether bilateral images are used at the same time.

**Justification Of Rating:**

The comparison study seems to be conducted in a hurry and results are incomplete. Also due to the high level of performance by previous publications, e.g. Wu et al., 2019 and McKinney et al., 2020, the study fails to improve the level of performance and therefore provides incremental value to the field.

**Paper Type:**

validation/application paper

**Special Issue:**

no

---

> ### Author Response · Authors · 2020-03-28
> **Response to Reviewer #1**
>
> We thank the reviewer for his time and the remarks.
>
> The main objection raised in the review was that the improvements were incremental compared to Wu et al. (2019) and McKinney et al. (2020). We respectfully disagree with this framing of our paper.
>
> The main objective of our work is improving utilization of multiple views in models for breast cancer screening exam classification. To study this question, we use the “Joint ResNet” model inspired by Wu et al. as a baseline. Our primary goal *was not* to improve the state-of-the-art performance, but to understand what makes multimodal learning difficult in the context of this task, and explore the efficacy of different (previously proposed) solutions.
>
> With this objective in mind, it is also worth noting that Wu et al.’s “image-and-heatmap” model trained without transfer learning from BIRADS achieved an AUC of 0.856, while the best performance achieved in our paper is 0.887. This is an over 20% reduction of error.
>
> For the specific question about the bilateral images, we treat each breast as an instance and do not differentiate between exams for left and right breasts in the training, validation and test phases. We clarified the description in the revised version of the paper.

---

> ### Comment · Area_Chair1 · 2020-03-28
> **Rebuttal**
>
> Dear Reviewer
>
> Can you read the rebuttal and see if it clarified the issues identified in your review?
>
> Thanks
> Your AC.

---

> > ### Comment · AnonReviewer1 · 2020-04-03
> > **rebuttal accepted**
> >
> > The reviewer agrees with the authors that the manuscript was successful in identifying and addressing the uneven progression problem in multi-modality training. The contribution is meaningful to problems in similar nature in the field of medical imaging.

---

### Meta-Review · Area_Chair1 · 2020-04-05
**MetaReview of Paper158 by AreaChair1**

**Rating:** 3
**Recommendation For Accepted Papers:** Poster

**Metareview:**

All reviewers acknowledge the importance of the paper, the fact that it is well written with clear hypotheses and good experiments, and meaningful solutions.  Reviews also listed a few issues, such as a limited number of lessons and conclusions reflected by a poor discussion section, and missing details about the experimental setup.  Rebuttal addressed well most of the issues identified.  I support the publication of the paper, but encourage the authors to address the issues identified.  In case the authors address the issues, my rating can be  between weak and strong accept and the paper can be either oral or poster.

**Paper Type:**

both

**Special Issue:**

yes

---

> ### Author Response · Authors · 2020-04-05
> **Response to the MetaReview**
>
> Thank you for your time and a positive review on the paper.
>
> We acknowledge the comments made by the reviewers. We will address them in the camera-ready version, including more experiments and an enriched discussion on the results.
>
> Best,
> Authors.

---

### Decision · Program_Chairs · 2020-04-11

Accept